# Efficient Spatially Sparse Inference for Conditional GANs and Diffusion Models

**Muyang Li**[1]  **Ji Lin**[2]  **Chenlin Meng**[3]  **Stefano Ermon**[3]  **Song Han**[2]  **Jun-Yan Zhu**[1]

[1]Carnegie Mellon University    [2]Massachusetts Institute of Technology    [3]Stanford University

## Abstract

During image editing, existing deep generative models tend to re-synthesize the entire output from scratch, including the unedited regions. This leads to a significant waste of computation, especially for minor editing operations. In this work, we present Spatially Sparse Inference (SSI), a general-purpose technique that selectively performs computation for edited regions and accelerates various generative models, including both conditional GANs and diffusion models. Our key observation is that users tend to make gradual changes to the input image. This motivates us to cache and reuse the feature maps of the original image. Given an edited image, we sparsely apply the convolutional filters to the edited regions while reusing the cached features for the unedited regions. Based on our algorithm, we further propose Sparse Incremental Generative Engine (SIGE) to convert the computation reduction to latency reduction on off-the-shelf hardware. With 1.2%-area edited regions, our method reduces the computation of DDIM by $7.5\times$ and GauGAN by $18\times$ while preserving the visual fidelity. With SIGE, we accelerate the inference time of DDIM by $3.0\times$ on RTX 3090 and $6.6\times$ on Apple M1 Pro CPU, and GauGAN by $4.2\times$ on RTX 3090 and $14\times$ on Apple M1 Pro CPU.

## 1   Introduction

Deep generative models, such as GANs [1, 2] and diffusion models [3, 4, 5], excel at synthesizing photo-realistic images, enabling many image synthesis and editing applications. For example, users can edit an image by drawing sketches [6, 7], semantic maps [6, 8], or strokes [9]. All of these applications require users to interact with generative models frequently and therefore demand fast inference time.

In practice, content creators often edit images gradually and only update a small image region each time. However, even for a minor edit, recent generative models often synthesize the entire image, including the unchanged regions, which leads to a significant waste of computation. As a concrete example shown in Figure 2(a), the result of the previous editing has already been computed, and the user further edits 9.4% area. However, the vanilla DDIM [5] needs to generate the entire image to obtain the newly edited regions, wasting 80% computation on the unchanged regions. A naive approach to address this issue would be to first segment the newly edited regions, synthesize the corresponding output regions, and blend the outputs back into the previous output. Unfortunately, this method often creates visible seams between the newly edited and unedited regions. How could we save the computation by only updating the edited regions without losing global coherence?

In this work, we propose Spatially Sparse Inference (SSI), a general method to accelerate deep generative models, including conditional GANs and diffusion models, by utilizing the spatial sparsity of edited regions. Our method is motivated by the observation that feature maps at the unedited regions remain mostly the same during user editing. As shown in Figure 2(b), our key idea is to reuse the cached feature maps of the previous editing and sparsely update the newly edited regions. Specifically, given user input, we first compute the difference mask to locate the newly edited regions. For each

36th Conference on Neural Information Processing Systems (NeurIPS 2022).

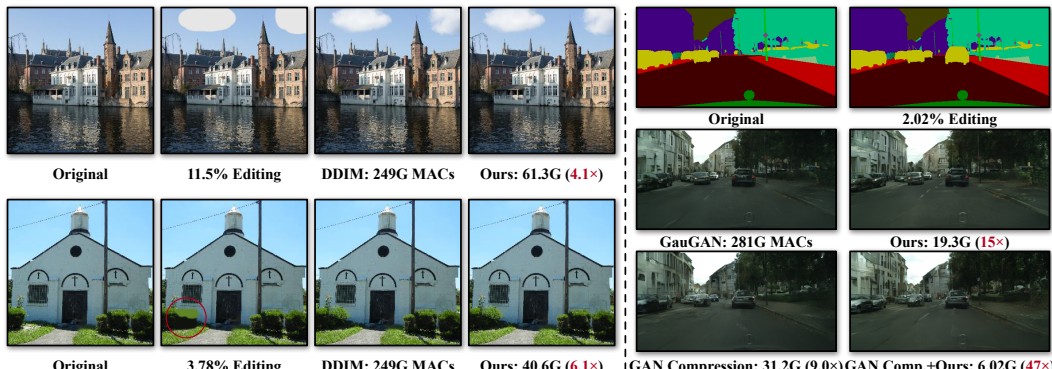

Figure 1: We introduce *Spatially Sparse Inference*, a general-purpose method to selectively perform computations at the edited regions for image editing applications. Our method reduces the computation of DDIM [5] by $4 \sim 6\times$ and GauGAN [8] by $15\times$ for the examples shown in the figures while preserving the image quality. When combined with existing model compression methods such as GAN Compression [10], our method further reduces the computation of GauGAN by $47\times$.

convolution layer in the model, we only apply the filters to the masked regions sparsely while reusing the previous activations for the unchanged regions. The sparse update can significantly reduce the computation without hurting the image quality. However, the sparse update involves a gather-scatter process, which often incurs significant latency overheads for existing deep learning frameworks. To address the issue, we propose *Sparse Incremental Generative Engine (SIGE)* to translate the theoretical computation reduction of our algorithm to measured latency reduction on various hardware.

To evaluate our method, we automatically create new image editing benchmark datasets on LSUN Church [11] and Cityscapes [12]. Without loss of visual fidelity, we reduce the computation of DDIM [5] by $7.5\times$, Progressive Distillation [13] by $2.7\times$, and GauGAN by $18\times$ measured by MACs*. Compared to existing generative model acceleration methods [10, 14, 15, 16, 17, 18, 19], our method directly uses the off-the-shelf pre-trained weights and could be applied to these methods as a plugin. When applied to GAN Compression [10], our method reduces the computation of GauGAN by $\sim 50\times$. See Figure 1 for some examples of our method. With SIGE, we accelerate DDIM $3.0\times$ on RTX 3090 GPU and $6.6\times$ on Apple M1 Pro CPU, and GauGAN $4.2\times$ on RTX 3090 GPU and $38\times$ on Apple M1 Pro CPU. Our code and benchmarks are available at https://github.com/lmxyy/sige.

## 2   Related Work

**Generative models.**   Generative models such as GANs [1, 2, 20, 21], diffusion models [4, 3, 22], and auto-regressive models [23, 24] have demonstrated impressive photorealistic synthesis capability. They have also been extended to conditional image synthesis tasks such as image-to-image translation [25, 6, 26, 27], controllable image generation [9, 28, 8], and real image editing [29, 28, 30, 31, 32, 33, 34, 27]. Unfortunately, recent generative models have become increasingly computationally intensive, compared to their recognition counterparts. For example, GauGAN [8] consumes 281G MACs, $500\times$ more than MobileNet [35, 36, 37]. Similarly, one key limitation of diffusion models [4] is their long inference time and substantial computation cost. To generate one image, DDPM requires hundreds or thousands of forwarding steps [4, 22], which is often infeasible in real-world interactive settings. To improve the sampling efficiency of DDPMs, recent works [5, 38, 39] propose to interpret the sampling process of DDPMs from the perspective of ordinary differential equations. However, these approaches still require hundreds of steps to generate high-quality samples. To further reduce the sampling cost, DDGAN [40] uses a multimodal conditional GAN to model each denoising step. Salimans *et al.* [13] propose to progressively distill a pre-trained DDPM model into a new model that requires fewer steps. Although this approach drastically reduces the sampling steps, the distilled model itself remains computationally prohibitive. Unlike prior work, our work focuses on reducing the computation cost of a pre-trained model. Our work is complementary to recent efforts on model compression, distillation, and the sampling step reduction of the diffusion models.

**Model acceleration.**   People apply model compression techniques, including pruning [41, 42, 43, 44, 45, 46] and quantization [41, 47, 48, 49, 50, 51], to reduce the computation and model size of off-the-shelf deep learning models. Recent works apply Neural Architecture Search (NAS) [52, 53, 54, 55,

---

*We measure the computational cost with the number of Multiply-Accumulate operations (MACs). 1 MAC=2 FLOPs.

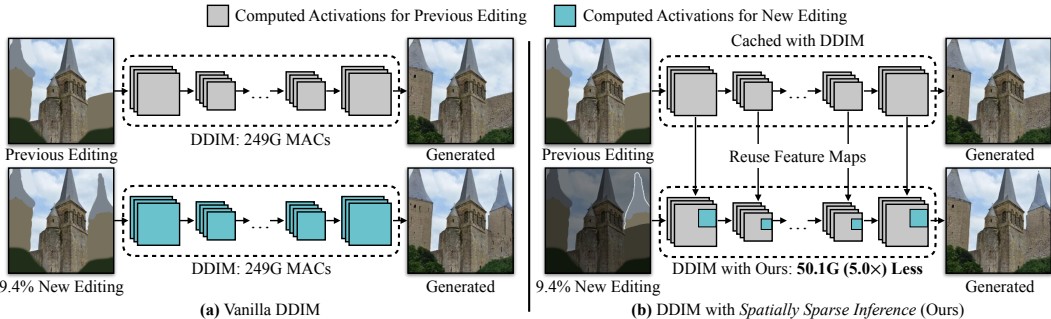

Figure 2: In the interactive editing scenario, a user edits 9.4% area. **(a)** Vanilla DDIM has to regenerate the *entire* image for every single editing, even though only 9.4% editing is made. **(b)** Our method instead *reuses* the feature maps of the previous editing and only applies convolutions to the *newly edited* regions sparsely, which has a $5.0\times$ MACs reduction in this case.

56, 57, 58] to automatically design efficient neural architectures. The above ideas can be successfully applied to accelerate the inference of GANs [10, 59, 60, 61, 14, 62, 15, 16, 17, 18, 19, 63]. Although these methods have achieved prominent compression and speedup ratios, they all reduce the computation from the model dimension but fail to exploit the redundancy in the spatial dimension during image editing. Besides, these methods require re-training the compressed model to maintain performance, while our method can be directly applied to existing pre-trained models. We show that our method can be combined with model compression [10] to achieve a $\sim 50\times$ MACs reduction in Section 4.1.

**Sparse computation.** Sparse computation has been widely explored in the weight domain [64, 65, 66, 67], input domain [68, 69], and activation domain [70, 71, 72, 73]. For activation sparsity, RRN [74] utilizes the sparsity in the consecutive video frame difference to accelerate video models. However, their sparsity is unstructured, which requires special hardware to reach its full speedup potential. Several works instead use structured sparsity. Li *et al.* [75] use a deep layer cascade to apply more convolutions on the hard regions than the easy regions to improve the accuracy and speed of semantic segmentation. To accelerate 3D object detection, SBNet [70] uses a spatial mask, either from a priori problem knowledge or an auxiliary network, to sparsify the activations. It adopts a tiling-based sparse convolution algorithm to handle spatial sparsity. Recent works further integrate the spatial mask generation network into the sparse inference network in an end-to-end manner [76] and extend the idea to different tasks [77, 78, 79, 80]. Compared to SBNet [70], our mask is directly derived from the difference between the original image and the edited image. Additionally, our method does not require any auxiliary network or extra model training. We also introduce other optimizations, such as normalization removal and kernel fusions, to better adapt our engine for image editing.

# 3 Method

We build our method based on the following observation: during interactive image editing, a user often only changes the image content gradually. As a result, only a small subset of pixels in a local region is being updated at any moment. Therefore, we can reuse the activations of the original image for the unedited regions. As shown in Figure 3, we first pre-compute all activations of the original input image. During the editing process, we locate the editing regions by computing a difference mask between the original and edited image. We then reuse the pre-computed activations for the unedited regions and only update the edited regions by applying convolutional filters to them. In Section 3.1, we show the sparsity in the intermediate activations and present our main algorithm. In Section 3.2, we discuss the technical details of how our Sparse Incremental Generative Engine (SIGE) supports the sparse inference and converts the theoretical computation reduction to measured speedup on hardware.

## 3.1 Activation Sparsity

**Preliminary.** First, we closely study the computation within a single layer. We denote $A_l^{\text{original}}$ and $A_l^{\text{edited}}$ as the input tensor of the original image and edited image to the $l$-th convolution layer $F_l$, respectively. $W_l$ and $b_l$ are the weight and bias of $F_l$. The output of $F_l$ with input $A_l^{\text{edited}}$ could be

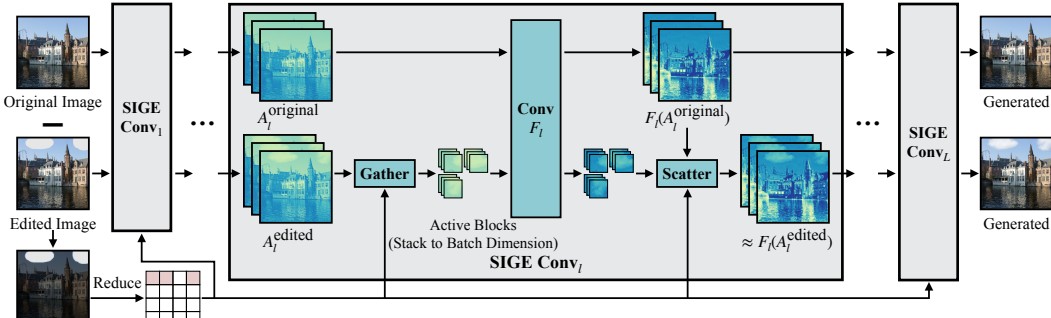

Figure 3: Tiling-based sparse convolution overview. For each convolution $F_l$ in the network, we wrap it into SIGE Conv$_l$. The activations of the original image are already pre-computed. When getting the edited image, we first compute a difference mask between the original and edited image and reduce the mask to the active block indices to locate the edited regions. In each SIGE Conv$_l$, we directly gather the active blocks from the edited activation $A_l^{\text{edited}}$ according to the reduced indices, stack the blocks along the batch dimension, and feed them into $F_l$. The gathered blocks have an overlap of width 2 if $F_l$ is $3 \times 3$ convolution [70]. After getting the output blocks from $F_l$, we scatter them back into $F_l(A_l^{\text{original}})$ to get the edited output, which approximates $F_l(A_l^{\text{edited}})$.

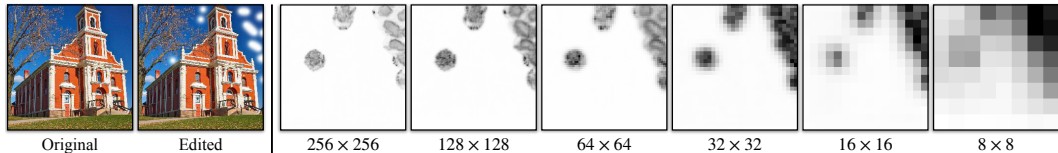

| Original | Edited | $256 \times 256$ | $128 \times 128$ | $64 \times 64$ | $32 \times 32$ | $16 \times 16$ | $8 \times 8$ |

Figure 4: Left: Detailed editing example. Right: Channel-wise average of $|\Delta A_l|$ at the $l$-th layer of DDIM with different feature map resolutions. $|\Delta A_l|$ is sparse and non-zero values are aggregated at the edited regions.

computed in the following way due to the linearity of convolution:

$$
\begin{aligned}
F_l(A_l^{\text{edited}}) &= W_l * A_l^{\text{edited}} + b_l \\
&= W_l * (A_l^{\text{edited}} - A_l^{\text{original}}) + (W_l * A_l^{\text{original}} + b_l) \\
&= W_l * \underbrace{\Delta A_l}_{\text{sparse}} + \underbrace{F_l(A_l^{\text{original}})}_{\text{pre-computed}},
\end{aligned}
$$

where $*$ is the convolution operator and $\Delta A_l = A_l^{\text{edited}} - A_l^{\text{original}}$. If we already pre-computed all the $F_l(A_l^{\text{original}})$, we only need to compute $W_l * \Delta A_l$. Naïvely, computing $W_l * \Delta A_l$ has the same complexity as $W_l * A_l^{\text{edited}}$. However, since the edited image shares similar features with the original image given a small edit, $\Delta A_l$ should be sparse. Below, we discuss different strategies to leverage the activation sparsity to accelerate model inference.

Our first attempt was to prune $\Delta A_l$ by zeroing out elements smaller than a certain threshold to achieve the target sparsity. Unfortunately, this pruning method fails to achieve measured speedup due to the overheads of the on-the-fly pruning and irregular sparsity pattern.

**Structured sparsity.** Fortunately, user edits are often highly structured and localized. As a result, $\Delta A_l$ should also share the *structured spatial sparsity*, where non-zero values are mostly aggregated within the edited regions, as shown in Figure 4. We then directly use the original image and edited image to compute a difference mask and sparsify $\Delta A_l$ with this mask.

## 3.2 Sparse Engine SIGE

But how could we leverage the structured sparsity to accelerate $W_l * \Delta A_l$? A naïve approach is to crop a rectangular edited region out of $\Delta A_l$ for each convolution and only compute features for the cropped regions. Unfortunately, this naïve cropping method works poorly for the irregular edited regions (*e.g.*, the example shown in Figure 4).

**Tiling-based sparse convolution.** Instead, as shown in Figure 5(a), we use a tiling-based sparse convolution algorithm. We first downsample the difference mask to different scales and dilate the downsampled masks with extra pixels (1 for diffusion models and 2 for GauGAN). Then we divide $\Delta A_l$ into multiple small blocks of the same size spatially and index the difference mask

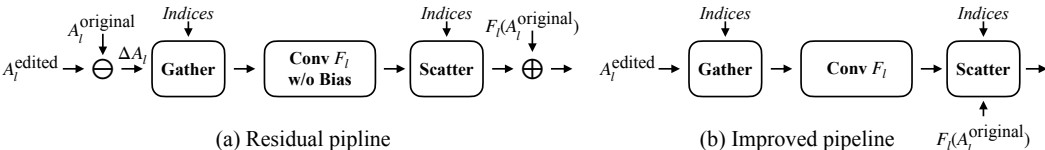

Figure 5: The titling-based sparse convolution pipelines. (a) We first compute the activation difference $\Delta A_l$ and gather the active blocks along the batch dimension from it according to the indices reduced from the difference mask. We then feed the blocks into the convolution $F_l$ without bias, scatter the output into a zero tensor, and add the residual $F_l(A_l^{\text{original}})$ back. (b) We directly gather the blocks from $A_l^{\text{edited}}$ without computing $\Delta A_l$. $F_l$ is computed with bias. We scatter the output into $F_l(A_l^{\text{original}})$ instead of a zero tensor.

at the corresponding resolution. Each block index refers to a single block with non-zero elements. We then gather the non-zero blocks (we also call them *active blocks*) accordingly along the batch dimension and feed them into the convolution $F_l$. Finally, we scatter the output blocks into a zero tensor according to the indices to recover the original spatial size and add the pre-computed residual $F_l(A^{\text{original}})$ back. The gathered active blocks have an overlap with width 2 for $3 \times 3$ convolution to ensure the output blocks of the adjacent input blocks are seamlessly stitched together [70].

This pipeline in Figure 5(a) is equivalent to a simpler pipeline in Figure 5(b). Instead of gathering $\Delta A_l$, we could directly gather $A_l^{\text{edited}}$. The convolution needs to be computed with bias $b_l$. Besides, we need to scatter the output blocks into $F_l(A_l^{\text{original}})$ instead of a zero tensor. Thus, we do not need to store $A_l^{\text{original}}$ anymore, which further saves memory and removes the overheads of addition and subtraction. Figure 3 visualizes the pipeline.

However, the aforementioned pipeline still fails to produce a noticeable speedup due to extra kernel calls and memory movement overheads in `Gather` and `Scatter`. For example, the original dense $3 \times 3$ convolution with 128 channels and input resolution $256 \times 256$ would take 0.78ms on RTX 3090. The sparse convolution using pipeline Figure 5(b) on the example shown in Figure 4 (15.5% edited regions) needs 0.42ms in total, while `Gather` and `Scatter` cost a significant overhead (0.17ms, about 41%). To reduce it, we further optimize SIGE by pre-computing normalization parameters and applying kernel fusion.

**Pre-computing normalization parameters.** For batch normalization [81], it is easy to remove the normalization layer during inference time since we can use pre-computed mean and variance statistics from model training. However, recent deep generative models often use instance normalization [82, 83] or group normalization [84, 85], which compute the statistics on the fly during inference. These normalization layers create overheads as we need to estimate the statistics from the full-size tensors. However, as the original and edited images are quite similar given a small user edit, we assume $A_l^{\text{original}} \approx A_l^{\text{edited}}$. This allows us to reuse the statistics of $A_l^{\text{original}}$ for the normalization instead of recomputing them for $A_l^{\text{edited}}$. Thus, normalization layers could be replaced by simple `Scale+Shift` operations with pre-computed $A_l^{\text{original}}$ statistics.

**Kernel fusion.** As mentioned before, both the `Gather` and `Scatter` operations introduce significant data movement overheads. To reduce it, we fuse several element-wise operations (`Scale+Shift` and `Nonlinearity`) into `Gather` and `Scatter` [70, 86, 87] and only apply these element-wise operations to the active blocks (*i.e.*, edited regions). Furthermore, we perform in-place computation to reduce the number of kernel calls and memory allocation overheads.

In `Scatter`, we need to copy the pre-computed activation $F_l(A_l^{\text{original}})$. This copying operation is highly redundant, as most elements from $F_l(A_l^{\text{original}})$ do not involve any computation given a small edit and will be discarded in the next `Gather`. To reduce the tensor copying overheads, we fuse the `Scatter` and the following `Gather` by directly gathering the active blocks from $F_l(A_l^{\text{original}})$ and the input blocks to be scattered. Sometimes, the residual connection in the ResBlock [88] contains shortcut $1 \times 1$ convolution to match the channel number of the residual and the ResBlock output. We also fuse the `Scatter` in the shortcut branch, main branch, and the residual addition together to avoid the tensor copying overheads in the shortcut `Scatter`. Please refer to Appendix A for more details.

# 4 Experiments

Below we first describe our models, baselines, datasets, and evaluation protocols. We then discuss our main qualitative and quantitative results. Finally, we include a detailed ablation study regarding the importance of each algorithmic design.

**Models.** We conduct experiments on three models, including diffusion models and GAN-based models, to explore the generality of our method.

- *DDIM* [5] is a fast sampling approach for diffusion models. It proposes to interpret the sampling process of diffusion models through the lens of ordinary differential equations.
- *Progressive Distillation (PD)* [13] adopts network distillation [89] to progressively reduce the number of steps for diffusion models.
- *GauGAN* [8] is a paired image-to-image translation model which learns to generate a high-fidelity image given a semantic label map.

**Baselines.** We compare our methods against the following baselines:

- *Patch*. We crop the smallest patch coverring all the edited regions, feed it into the model, and blend the output patch into the original image.
- *Crop*. For each convolution $F_l$, we crop the smallest rectangular region that covers all masked elements of the activation $A_l^{\text{edited}}$, feed it into $F_l$, and scatter the output patch into $F_l(A_l^{\text{original}})$.
- *40% Pruning*. We uniformly prune 40% weights of the models without further fine-tuning, as our method directly uses the pre-trained weights. Since the fine-grained pruning is unstructured, it requires special hardware to achieve measured speedup, so we do not report MACs for this baseline.
- *0.19 GauGAN*. We reduce each convolution layer of GauGAN to $19\%$ channels ($21\times$ MACs reduction) and train it from scratch.
- *GAN Compression* [10]. A general-purpose compression method for conditional GANs. *GAN Comp. (S)* means GAN Compression with a larger compression ratio.
- *0.5 Original* means linearly scaling each layer of the original model to 50% channels, and we only use this to benchmark our efficiency results.

**Datasets.** We use the following two datasets in our experiments:

- *LSUN Church*. We use the LSUN Church Outdoor dataset [11] and follow the same preprocessing steps as prior works [4, 38]. To automatically generate a stroke editing benchmark, we first use Detic [90] to segment the images in the validation set. For each segmented object, we use its segmentation mask to inpaint the image by CoModGAN [91] and treat the inpainted image as the original image. We generate the corresponding user strokes by first blurring the masked regions with the median filter and quantizing it into 6 colors following SDEdit [9]. We collect 454 editing pairs in total (431 synthetic + 23 manual). We evaluate DDIM [5] and PD [13] on this dataset.
- *Cityscapes*. The dataset [12] contains images of German street scenes. The training and validation sets consist of 2975 and 500 images, respectively. Our editing dataset has 1505 editing pairs in total. We evaluate GauGAN [8] on this dataset.

Please refer to Appendix B for more details about the benchmark datasets.

**Metrics.** Following previous works [9, 10, 8], we use the standard metrics Peak Signal Noise Ratio (PSNR, higher is better), LPIPS (lower is better) [92], and Fréchet Inception Distance (FID, lower is better) [93, 94][†] to evaluate the image quality on both LSUN Church [11] and Cityscapes [12]. For Cityscapes, we adopt a semantic segmentation metric to evaluate the generated images. Specifically, we run DRN-D-105 [95] on the generated images and compute the mean Intersection over Union (mIoU) of the segmentation results. Generally, a higher mIOU indicates that the generated images look more realistic and better align to the input.

---

[†]We use clean-fid for FID calculation.

| Model | Method | MACs | | PSNR (↑) | | LPIPS (↓) | | FID (↓) | mIoU (↑) |
|---|---|---|---|---|---|---|---|---|---|
| | | Value | Ratio | with G.T. | with Orig. | with G.T. | with Orig. | | |
| DDIM | Original | 249G | – | 26.8 | – | 0.069 | – | 65.4 | – |
| | 40% Pruning | – | – | 24.9 | 31.0 | 0.991 | 0.101 | 72.2 | – |
| | Patch | 72.0G | 3.5× | 26.8 | 40.6 | 0.076 | 0.022 | 66.4 | – |
| | **Ours** | **65.3G** | **3.8×** | **26.8** | **53.4** | **0.070** | **0.009** | 65.8 | – |
| PD | Original | 66.9G | – | 21.9 | – | 0.182 | – | 90.0 | – |
| | 40% Pruning | – | – | 21.6 | 37.6 | 0.189 | 0.051 | 101 | – |
| | **Ours** | **32.5G** | **2.1×** | **21.9** | **60.7** | **0.183** | **0.003** | 90.1 | – |
| GauGAN | Original | 281G | – | 15.8 | – | 0.451 | – | 55.2 | 62.4 |
| | GAN Comp. [10] | 31.2G | 9.0× | 15.7 | 19.5 | **0.453** | 0.282 | 55.3 | 61.5 |
| | **Ours** | **30.7G** | **9.2×** | **15.8** | **26.5** | 0.454 | **0.113** | **54.2** | **62.1** |
| | 0.19 GauGAN | 13.3G | 21× | 15.5 | 18.6 | 0.462 | 0.319 | 57.8 | 53.5 |
| | GAN Comp. (S) | 9.64G | 29× | 15.7 | 19.1 | 0.461 | 0.306 | **50.3** | 57.4 |
| | **GAN Comp.+Ours** | **7.06G** | **40×** | **15.7** | **19.1** | **0.457** | **0.294** | 54.4 | **60.1** |

Table 1: Quantitative quality evaluation. PSNR/LPIPS *with G.T.* means computing the metrics with the ground-truth images, and *with Orig.* means computing with the generated samples from the original model. *40% Pruning*: Uniformly pruning 40% weights of the model without fine-tuning. *Patch*: Cropping the smallest image patch that covers all the edited region and blending the output patch into the original image. *0.19 GauGAN*: Uniformly reducing each layer of GauGAN to 19% channels and training from scratch. *GAN Comp. (S)*: GAN Compression with a larger compression ratio. For all models, our method outperforms other baselines with less computation. When applying our method to GAN Compression, we reduce the MACs of GauGAN by 40× with minor performance degradation.

**Implementation details.** The number of denoising steps for DDIM and PD are 100 and 8, respectively, and we use 50 and 5 steps for SDEdit. We dilate the difference mask by 5, 2, 5, and 1 pixels for DDIM, PD with resolution 128, PD with resolution 256 and GauGAN, respectively. Besides, we apply our sparse kernel to all convolution layers whose input feature map resolutions are larger than $32 \times 32$, $16 \times 16$, $8 \times 16$ and $16 \times 32$ for DDIM, PD, original GauGAN and GAN Compression, respectively. For DDIM [5] and PD [13], we pre-compute and reuse the statistics of the original image for all group normalization layers [84]. For GAN Compression [10], we pre-compute and reuse the statistics of the original image for all instance normalization layers [82] whose resolution is higher than $16 \times 32$. For all models, the sparse block size for $3 \times 3$ convolution is 6 and $1 \times 1$ convolution is 4.

## 4.1 Main Results

**Image quality.** We report the quantitative results of applying our method on DDIM [5], Progressive Distillation (PD) [13], and GauGAN [8] in Table 1 and show the qualitative results in Figure 6. For PSNR and LPIPS, *with G.T.* means computing the metric with the ground-truth images. *With Orig.* means computing the metric with the samples generated by the original model. On LSUN Church, we only use 431 synthetic images for the *PSNR/LPIPS with G.T.* metrics, as manual editing does not have ground truths. For the other metrics, we use the entire LSUN Church dataset (431 synthetic + 23 manual). On Cityscapes, we view the synthetic semantic maps as the original input and the ground-truth semantic maps as the edited input for the *PSNR/LPIPS with G.T.* metrics, which has 1505 samples. For the other metrics, we include symmetric editing (view the ground-truth semantic maps as the original inputs and synthetic semantic maps as the edited inputs), which has 3010 samples in total. For the models with method *Patch* and *Ours*, whose computation is editing dependent, we measure the average MACs over the whole dataset.

For DDIM and Progressive Distillation, our method outperforms all baselines consistently and achieves results on par with the original model. The *Patch* inference fails when the edited region is small as the global context is insufficient. Although our method only applies convolutional filters to the local edited regions, we could reuse the global context stored in the original activations. Therefore, our method could perform the same as the original model. For GauGAN, our method also performs better than GAN Compression [10] with an even larger MACs reduction. When applying our method to GAN Compression, we further achieve a $\sim 40\times$ MACs reduction with minor performance degradation, beating both *0.19 GauGAN* and *GAN Comp. (S)*.

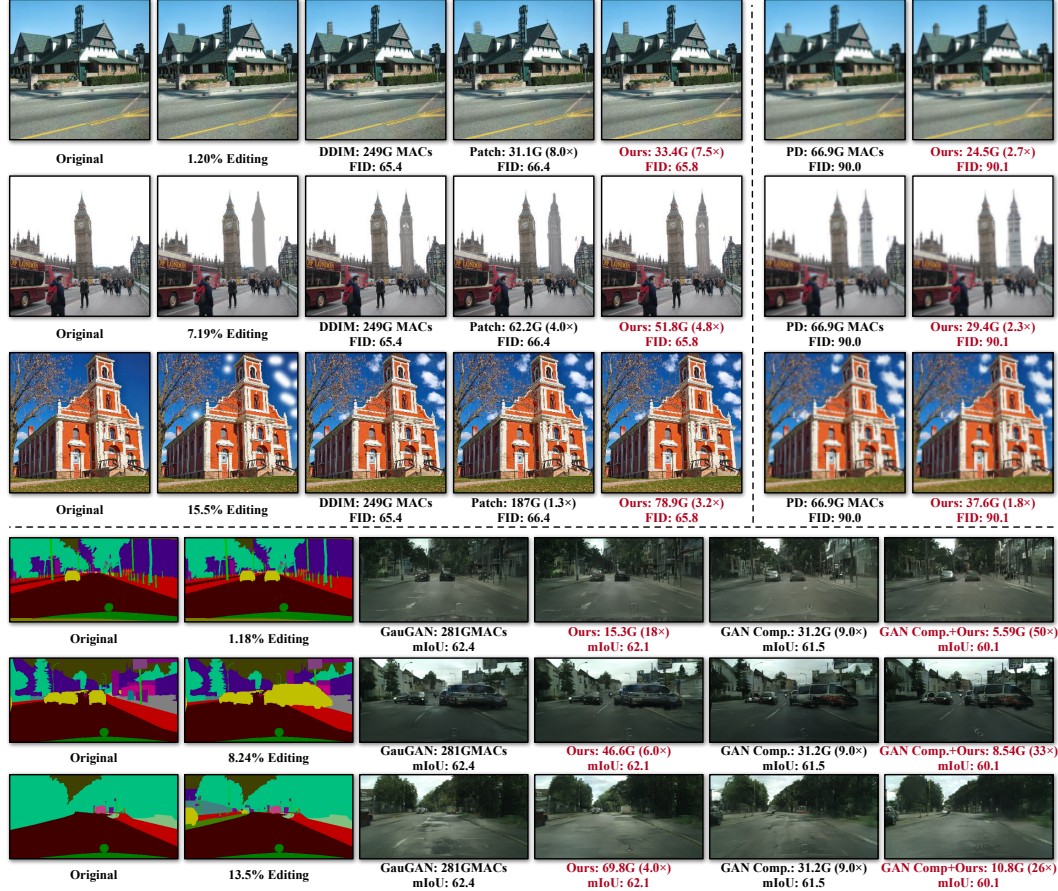

Figure 6: Qualitative results of our method under different editing sizes. Our method well preserves the visual fidelity of the original model without losing global context. On the contrast, *Patch* (cropping the smallest image patch that covers all the edited regions and scatter the output patch back into the original image) performs poorly because of the lack of global context when the editing is small.

**Model efficiency.** For real-world interactive image editing applications, inference acceleration on hardware is more critical than the computation reduction. To verify the effectiveness of our proposed engine, we measure the speedup of the editing examples shown in Figure 6 on four devices, including NVIDIA RTX 3090, NVIDIA RTX 2080Ti, Intel Core i9-10920X CPU, and Apple M1 Pro CPU, with different computational powers. We use batch size 1 to simulate real-world use. For GPU devices, we first perform 200 warm-up runs and measure the average latency of the next 200 runs. For CPU devices, we perform 10 warm-up runs and 10 test runs, repeat this process 5 times and report the average latency. The latency is measured in PyTorch 1.7[‡]. The results are shown in Table 2.

The original Progressive Distillation [13] can only generate $128 \times 128$ images, which is too small for real use. We add some extra layers to adapt the model to resolution $256 \times 256$. For fair comparisons, we also pre-compute the normalization parameters for the *Crop* baseline. When the editing pattern is like a rectangle, this baseline reduce similar computation with ours (*e.g.*, the first example of DDIM in Figure 6). However, the speedup is still worse than ours on RTX 3090, 2080Ti and Intel Core i9-10920X due to the large memory index overheads in native PyTorch. When the edited region is far from a rectangle (*e.g.*, the third example of DDIM), the cropped patches have much redundancy. Therefore, even though only 15.5% region is edited, the MACs reduction is only 1.6×. With 1.2% editing size, our method achieves a 7.5×, 4.6×, and 18× MACs reduction for DDIM, Progressive Distillation, and GauGAN, respectively. With SIGE, we achieve at most 4.1×, 2.9×, 6.0× and 14× speedup on RTX 3090, 2080Ti, Intel Core i9-10920X and Apple M1 Pro CPU, respectively. When applied to GAN Compression, SIGE achieves a 9.5× and 38× latency reduction on Intel Core i9-10920X and Apple M1 Pro CPU, respectively.

---

[‡] https://github.com/pytorch/pytorch

| Model | Editing Size | Method | MACs Value | MACs Ratio | 3090 Value | 3090 Ratio | 2080Ti Value | 2080Ti Ratio | Intel Core i9-10920X Value | Intel Core i9-10920X Ratio | Apple M1 Pro Value | Apple M1 Pro Ratio |
|---|---|---|---|---|---|---|---|---|---|---|---|---|
| DDIM | – | Original | 248G | – | 37.5ms | – | 54.6ms | – | 609ms | – | 12.9s | – |
| | | 0.5 Original | 62.5G | 4.0× | 20.0ms | 1.9× | 31.2ms | 1.8× | 215ms | 2.8× | 3.22s | 4.0× |
| | 1.20% | Crop | **32.6G** | **7.6×** | 15.5ms | 2.4× | 29.3ms | 1.9× | 185ms | 3.3× | **1.85s** | **6.9×** |
| | | Ours | 33.4G | 7.5× | **12.6ms** | **3.0×** | **19.1ms** | **2.9×** | **147ms** | **4.1×** | 1.96s | 6.6× |
| | 15.5% | Crop | 155G | 1.6× | 30.5ms | 1.2× | 44.5ms | 1.2× | 441ms | 1.4× | 8.09s | 1.6× |
| | | Ours | 78.9G | 3.2× | 19.4ms | 1.9× | 29.8ms | 1.8× | 304ms | 2.0× | 5.04s | 2.6× |
| PD256 | – | Original | 119G | – | 35.1ms | – | 51.2ms | – | 388ms | – | 6.18s | – |
| | | 0.5 Original | 31.0G | 3.8× | 29.4ms | 1.2× | 43.2ms | 1.2× | 186ms | 2.1× | 1.72s | 3.6× |
| | 1.20% | Ours | **25.9G** | **4.6×** | 18.6ms | 1.9× | 26.4ms | 1.9× | 152ms | 2.5× | 1.55s | 4.0× |
| | 15.5% | Ours | 48.5G | 2.5× | 21.4ms | 1.6× | 30.7ms | 1.7× | 250ms | 1.6× | 3.22s | 1.9× |
| GauGAN | – | Original | 281G | – | 45.4ms | – | 49.5ms | – | 682ms | – | 14.1s | – |
| | | GAN Compression | 31.2G | 9.0× | 17.0ms | 2.7× | 25.0ms | 2.0× | 333ms | 2.1× | 2.11s | 6.7× |
| | 1.18% | Ours | **15.3G** | **18×** | **11.1ms** | **4.1×** | **19.3ms** | **2.6×** | **114ms** | **6.0×** | **0.990s** | **14×** |
| | | GAN Comp.+Ours | **5.59G** | **50×** | **10.8ms** | **4.2×** | **16.2ms** | **3.1×** | **53.1ms** | **13×** | **0.370s** | **38×** |
| | 13.5% | Ours | 69.8G | 4.0× | 17.8ms | 2.5× | 27.1ms | 1.8× | 238ms | 2.9× | 4.06s | 3.5× |
| | | GAN Comp.+Ours | 10.8G | 26× | 11.8ms | 3.8× | 17.4ms | 2.8× | 94.4ms | 7.2× | 0.741s | 19× |

Table 2: Measured latency speedup on different devices. The detailed editing examples are shown in Figure 6. *0.5 Original*: Linearly scaling each layer of the model to 50% channels. *Crop*: For each convolution, we find a smallest patch covering the masked elements, crop it out, feed it into the convolution and scatter the output patch into the original image activation. Our method could reduce up to 18× MACs, and achieve up to 4.1×, 2.9×, 6.0× and 14× latency reduction on NVIDIA RTX 3090, 2080Ti, Intel Core i9-10920X and M1 Pro CPU. With GAN Compression, we could further speed up GauGAN by 9.5× on Intel Core-i9 and 38× on Apple M1 Pro CPU.

| MACs | Optimizations Sparse | Norm. | Elem. | Sct. | Latency Value | Ratio |
|---|---|---|---|---|---|---|
| 249G | | | | | 54.6ms | – |
| | ✓ | | | | 34.0ms | 1.6× |
| 32.6G | ✓ | ✓ | | | 29.6ms | 1.8× |
| (7.6×) | ✓ | ✓ | ✓ | | 20.7ms | 2.6× |
| | ✓ | ✓ | ✓ | ✓ | 19.1ms | 2.9× |

(a)

| Method | Editing Size | MACs Value | MACs Ratio | PyTorch Value | PyTorch Ratio | TensorRT Value | TensorRT Ratio |
|---|---|---|---|---|---|---|---|
| Original | – | 249G | – | 54.6ms | – | 47.7ms | – |
| Ours | 1.20% | 33.4G | 7.5× | 19.1ms | 2.9× | 14.4ms | **3.3×** |
| | 7.19% | 51.8G | 4.8× | 22.1ms | 2.5× | 18.6ms | **2.6×** |
| | 15.5% | 78.9G | 3.2× | 29.8ms | **1.8×** | 26.9ms | **1.8×** |

(b)

Table 3: (a) Ablation study of each kernel optimization. **Sparse**: Using tiling-based sparse convolution. **Norm.**: Pre-computing normalization parameters. **Elem.**: Fusing element-wise operations. **Sct.**: Fusing `Scatter` to reduce the tensor copying overheads. With all optimizations, we could reduce the latency of DDIM by 2.9× on NVIDIA RTX 2080Ti. (b) Latency comparisons of DDIM on RTX 2080Ti between PyTorch and TensorRT. The speedup ratio is larger in TensorRT than PyTorch, especially when the editing size is small.

## 4.2 Ablation Study

Below we perform several ablation studies to show the effectiveness of each design choice.

**Memory usage.** The pre-computed activations of the original image require additional memory storage. We profile the peak memory usage of the original model and our method in PyTorch. Our method only increases the peak memory usage of a single forward for DDIM, PD, GauGAN, and GAN Compression by 0.1G, 0.1G, 0.8G, and 0.3G, respectively. Specifically, our method needs to store additional 169M, 56M, 275M, and 120M parameters for DDIM [5], PD [13], original GauGAN [8] and GAN Compression [10], respectively, for a single forward. For the diffusion models, we need to store activations for all iteration steps (*e.g.*, 50 for DDIM and 5 for PD). However, data movement and kernel computation are asynchronized on GPU, so we could store the activations in CPU memory and load the on-demand ones on GPU to reduce peak memory usage.

**Speedup of each design.** Table 3a shows the effectiveness of each kernel optimization we add to SIGE for DDIM [5] on RTX 2080Ti. Naïvely applying the tiling-based sparse convolution could reduce the computation by 7.6×. Still, the latency reduction is only 1.6× due to the large memory overheads in `Gather` and `Scatter`. Pre-computing the normalization parameters could remove the latency of normalization statistics calculation and reduce the overall latency to 29.6ms. Fusing

element-wise operations into the `Gather` and `Scatter` could remove some redundant operations that are applied to the unedited regions and also reduce the memory allocation overheads (about 9ms). Finally, fusing the `Scatter` and `Gather` to `Scatter-Gather` and `Scatter` in the shortcut branch and main branch could further reduce about 1.6ms tensor copying overheads, achieving 2.9× speedup.

**Experiments with TensorRT.** Real-world model deployment also depends on deep learning backends with optimized libraries and runtimes. To demonstrate the effectiveness and extensibility of SIGE, we also implement our kernels in a widely-used backend TensorRT[§] and benchmark the DDIM latency results on RTX 2080Ti in Table 3b. Specifically, our speedup ratio becomes more prominent with TensorRT compared to PyTorch, especially for small edits, as TensorRT better supports small convolutional kernels with higher GPU utilization than PyTorch.

## 5   Conclusion & Discussion

For image editing, existing deep generative models often waste computation by re-synthesizing the image regions that do not require modifications. To solve this issue, we have presented a general-purpose method, Spatially Sparse Inference (SSI), to selectively perform computation on edited regions, and Sparse Incremental Generative Engine (SIGE) to convert the computation reduction to latency reduction on commonly-used hardware. We have demonstrated the effectiveness of our approach in various hardware settings.

**Limitations.** As discussed in Section 4.2, our method requires extra memory to store the original activations, which slightly increases the peak GPU memory usage. It may not work on certain memory-constrained devices, especially for the diffusion models (*e.g.*, DDIM [5]), since our method requires storing activations of all denoising steps.

Our engine has limited speedup on convolution with low resolution. When the input resolution is low, the active block size needs to be even smaller to get a decent sparsity, such as 1 or 2. However, such extremely small block sizes have bad memory locality and will result in low hardware efficiency.

Besides, we sometimes observe noticeable boundary between the edited regions and unedited regions in our generated samples of GauGAN [8]. This is because, for GauGAN model, the unedited region will also change slightly when we perform normal inference. However, since our method does not update the unedited region, there may be some visible seams between the edited and unedited regions, even though the semantic is coherent. Dilating the difference mask would help reduce the gap.

In most cases, the edit will only update the edited regions. However, sometimes the edit will also introduce global illumination changes such as shadow and reflection. For this case, as we only update the edited regions, we cannot update the global changes outside the edited regions accordingly.

**Societal impact.** In this paper, we investigate how to update user editing locally without losing global coherence to enable smoother interaction with the generative models. In real-world scenarios, people could use an interactive interface to edit an image, and our method could provide a quick and high-quality preview for their editing, which eases the process of visual content creation and reduces energy consumption, leading to a greener AI application. The reduced cost also provides a good user experience for lower-end devices, which further democratize the applications of generative models.

However, our method can be utilized by malicious users to generate fake contents, deceive people, and spread misinformation, which may lead to potential negative social impacts. Following previous works [9], we explicitly specify the usage permission of our engine with proper licenses. Additionally, we run a forensics detector [96] to detect the generated results of our method. On GauGAN, our generated images can be detected with $97.2\%$ average precision (AP). However, on DDIM [5] and Progressive Distillation [13], the APs are only $56.6\%$ and $52.4\%$. Such low APs are caused by the model differences between GANs and diffusion models, as observed in SDEdit [9]. We believe developing forensic methods for diffusion models is a critical future research direction.

**Acknowledgment.** We thank Yaoyao Ding, Zihao Ye, Lianmin Zheng, Haotian Tang, and Ligeng Zhu for the helpful comments on the engine design. We also thank George Cazenavette, Kangle Deng, Ruihan Gao, Daohan Lu, Sheng-Yu Wang and Bingliang Zhang for their valuable feedback. The project is partly supported by NSF, MIT-IBM Watson AI Lab, Kwai Inc, and Sony Corporation.

---

[§]We benchmark the results with TensorRT 8.4.

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
