# Spatially Sparse Inference for Generative Image Editing Supplementary Material

## A Additional Implementation Details

For all models, we use block size 6 for $3 \times 3$ convolutions and block size 4 for $1 \times 1$ convolutions. For DDIM [1] and Progressive Distillation [12], we pre-compute and reuse the statistics of the original image for all group normalization layers [84]. For GAN Compression [3], we pre-compute and reuse the statistics of the original image for all instance normalization layers [82] whose resolution is higher than $16 \times 32$.

## B Kernel Fusion

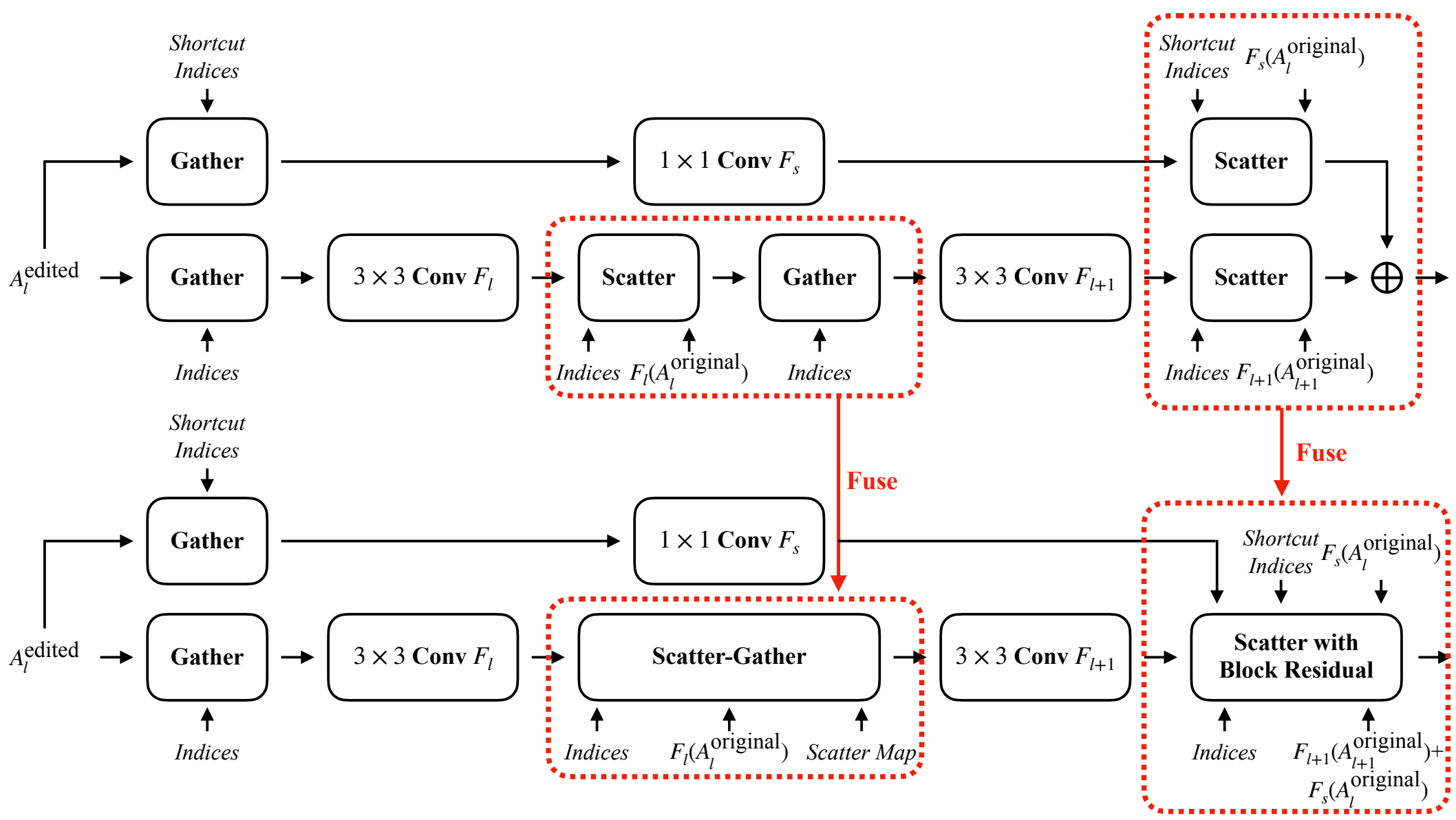

Figure 6: Visualization of kernel fusion in DDIM [1] ResBlock [88]. We omit the element-wise operations for simplicity and follow the notations in Section 3. As the kernel sizes of the convolution in the shortcut branch and main branch are different, their reduced active block indices are different (*Indices* and *Shortcut Indices*). To reduce the tensor copying overheads in `Scatter`, we fuse `Scatter` and the following `Gather` into `Scatter-Gather` and fuse the `Scatter` in the shortcut, main branch and residual addition into `Scatter with Block Residual`. We pre-compute an additional *Scatter Map* for the `Scatter-Gather` kernel.

As mentioned in Section 3.2, we fuse `Scatter` and the following `Gather` into a `Scatter-Gather` operator and also fuse `Scatter` in the shortcut, main branch and residual addition together. The detailed fusion pattern is shown in Figure 6. For simplicity, we omit the element-wise operations (*e.g.*, `Nonlinearity` and `Scale+Shift`). Below we include more implementation details of each fusion design.

**Scatter-Gather fusion.** When a `Scatter` is directly followed by a `Gather`, we could fuse these two operators into a `Scatter-Gather` to avoid copying the original activation $F_l(A_l^{\text{original}})$. We pre-built a *Scatter Map* to indicate the index mapping from the $F_l$ output to the previous `Scatter` output, and directly gather the active blocks from the $F_l$ output and original activation $F_l(A_l^{\text{original}})$ with it. Note that the pre-computation is cheap and only needs to be once for each resolution.

**Shortcut Scatter fusion.** The $1 \times 1$ convolution in the shortcut branch consumes much less computation than the convolutions in the main branch, therefore the overheads of `Gather` and `Scatter` weigh more in the shortcut branch. We fuse the `Scatter` in the shortcut branch and main branch along with residual addition together into `Scatter with Block Residual` to reduce these overheads. Specifically, we first scatter $F_{l+1}$ output in the pre-computed $F_{l+1}(A_l^{\text{original}}) + F_s(A_l^{\text{original}})$

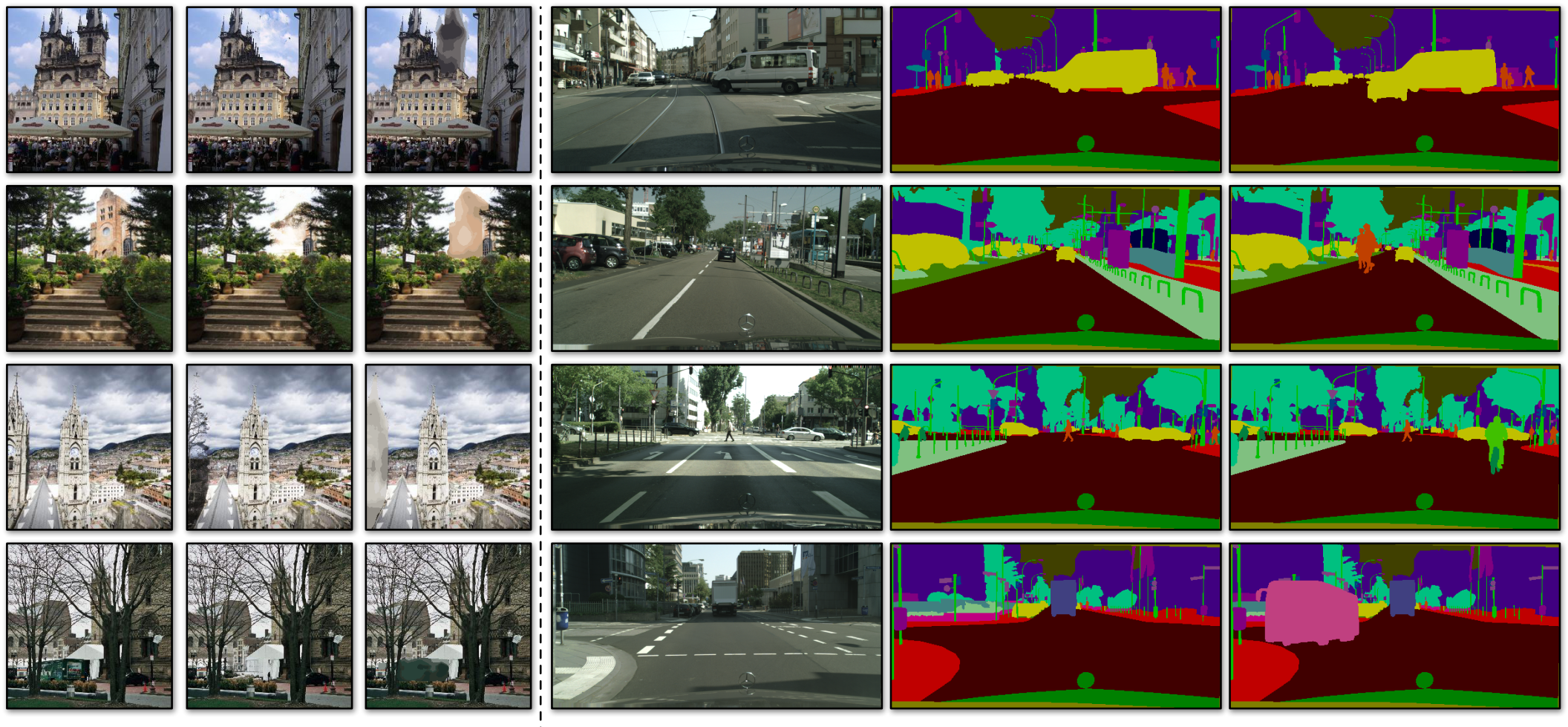

(a) LSUN-Church (b) Cityscapes

Figure 7: Several examples of our synthetic editing dataset on (a) LSUN Church and (b) Cityscapes. On LSUN Church, we view the inpainted image as the original image and generate the editing by quantizing color at the corresponding regions. On Cityscapes, we generate the editing by pasting some foreground objects to the ground-truth semantic maps.

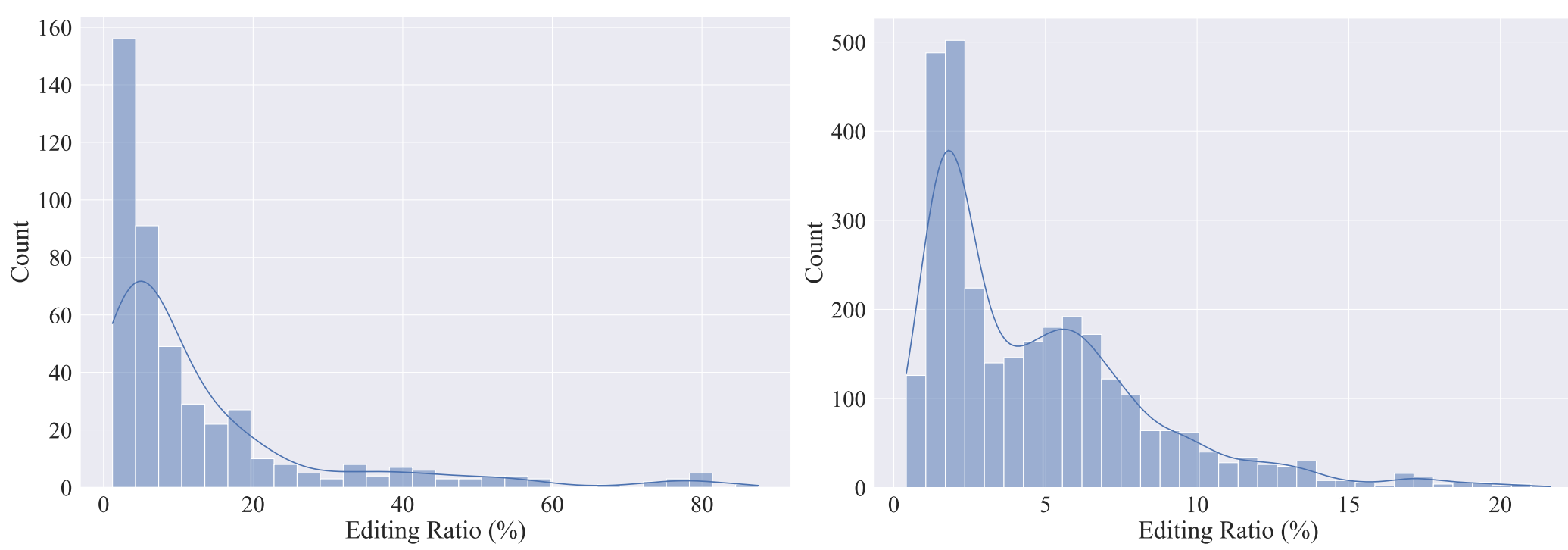

(a) LSUN Church. (b) Cityscapes.

Figure 8: Detailed editing ratio distribution of our synthetic datasets.

and add the original residual $F_s(A_l^{\text{original}})$ only at the scattered locations correspondingly according to *Indices*. Then we calibrate the final output with $F_s$ output by adding the residual difference $F_s(A_l^{\text{edited}}) - F_s(A_l^{\text{original}})$ at the scattered locations inplace according to *Shortcut Indices*.

# C Benchmark Datasets

We elaborate more details on how we build the synthetic editing dataset.

**LSUN Church.** Figure 7(a) shows some examples of our synthetic editing on LSUN Church. The average edited area of the whole dataset is 13.1%. The detailed distribution is shown in Figure 8a.

**Cityscapes.** We collect 27 foreground object semantic masks from the validation set. The objects include 4 bicycles, 1 motorcycle, 7 cars, 6 trucks, 3 buses, 5 persons, and 1 train. Figure 9 shows some visualization of the collected semantic masks. We generate the editing by randomly pasting one of these objects to the ground-truth semantic maps with augmentation. The augmentation includes random horizontal flip, resize (scale factor in $[0.8, 1.2]$), translation ($[-32, 32]$ for height and $[-64, 64]$ for width). To make the synthetic editing more reasonable, when the scale factor is larger than 1, the height translation can only be positive, otherwise, it can only be negative. Figure 7(b)

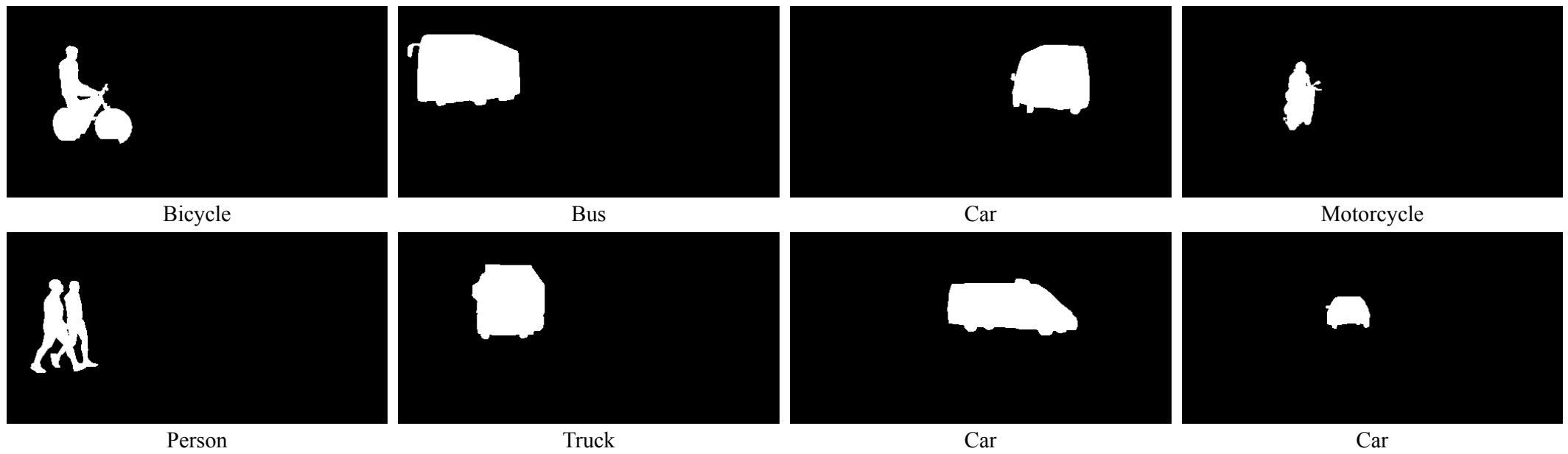

Figure 9: Several examples of our collected foreground object semantic masks.

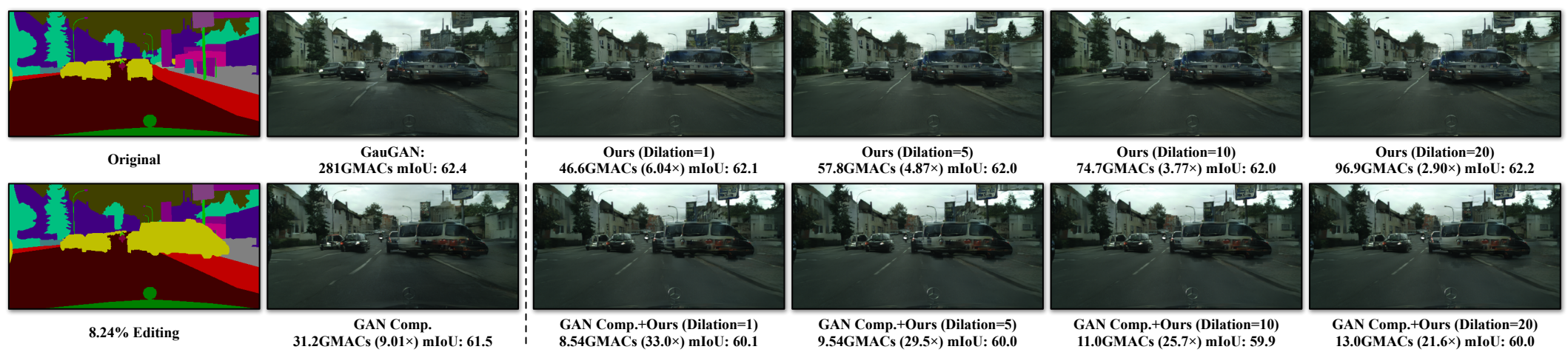

Figure 10: Visualization results of different dilation sizes on GauGAN. Although without mIoU improvement, increasing the dilation could smoothly blend the boundary between the edited region and unedited regions to improve the image quality slightly. Specifically, the shadow boundary of the added car fades when dilation increases. However, it will incur more computations.

shows some editing examples. The average editing area of the entire dataset is 4.77%. The detailed distribution is shown in Figure 8b.

## D Additional Results

**Dilation hyper-parameter.** We show the results of our method with different dilation sizes on GauGAN in Figure 10. Increasing the dilation brings more computations but also slightly improves the image quality. Specifically, the shadow boundary of the added car fades when increasing the dilation. We choose dilation 1 as the image quality is almost the same as 20 while delivering the best speed.

**Large editing.** In Table 4 and Figure 11, we show the results of large editing ($\sim 35\%$) using our method. Specifically, we could achieve at most $1.7\times$ speedup on DDIM, $1.5\times$ speedup on PD256 and $1.7\times$ speedup on GauGAN without losing visual fidelity. Furthermore, in many practical cases, users can decompose a large edit into several small edits. Our method could incrementally update the results instantly when the edit is being created.

**Sequential editing.** In Figure 12, we show the results of sequential editing with our method. Specifically, *One-time Pre-computation* performs as well as the *Full Model*, demonstrating that our method can be applied to multiple sequential editing with only one-time pre-computation in most cases. Moreover, for extremely large edited regions, we could use SIGE to incrementally update the pre-computed features (*Incremental Pre-computation*) and condition the later editing on the recomputed one. Its results are also as good as the full model. Therefore, our method could well address the sequential editing.

**Additional visualization.** In Figure 13, we show additional synthetic editing visual results of DDIM [1] and Progressive Distillation [12] on LSUN Church [10]. In Figure 14, we show additional synthetic editing visual results of GauGAN on Cityscapes [11].

## E License & Computation Resources

Here we show all the licenses of our used assets. The model DDIM [1], Progressive Distillation [12], GauGAN [2] and GAN Compression [3] is under MIT license, Apache license, Creative Commons license and BSD license, respectively. SDEdit is under MIT license. The license of Cityscapes [11] is here. LSUN Church [10] does not have explicit license.

| Model | Editing Size | Method | MACs | | 3090 | | 2080Ti | | Intel Core i9-10920X | | Apple M1 Pro | |
|---|---|---|---|---|---|---|---|---|---|---|---|---|
| | | | Value | Ratio | Value | Ratio | Value | Ratio | Value | Ratio | Value | Ratio |
| DDIM | – | Original | 248G | – | 37.5ms | – | 54.6ms | – | 609ms | – | 12.9s | – |
| | 32.9% | Ours | 115G | 2.2× | 26.0ms | 1.4× | 36.9ms | 1.5× | 449ms | 1.4× | 7.53s | 1.7× |
| PD256 | – | Original | 119G | – | 35.1ms | – | 51.2ms | – | 388ms | – | 6.18s | – |
| | 32.9% | Ours | 64.3G | 1.9× | 25.3ms | 1.4× | 35.1ms | 1.5× | 334ms | 1.2× | 4.47s | 1.4× |
| GauGAN | – | Original | 281G | – | 45.4ms | – | 49.5ms | – | 682ms | – | 14.1s | – |
| | | GAN Compression | 31.2G | 9.0× | 17.0ms | 2.7× | 25.0ms | 2.0× | 333ms | 2.1× | 2.11s | 6.7× |
| | 38.7% | Ours | 148G | 1.9× | 27.9ms | 1.6× | 41.7ms | 1.2× | 512ms | 1.3× | 8.37s | 1.7× |
| | | GAN Comp.+Ours | 18.3G | 15× | 15.3ms | 3.0× | 22.2ms | 2.2× | 169ms | 4.0× | 1.25s | 11× |

Table 4: Measured latency speedup of large editing on different devices. The detailed editing examples are shown in Figure 11. Our method could reduce up to 2.2× MACs, and 1.4×, 1.5×, 1.4× and 1.7× latency on NVIDIA RTX 3090, 2080Ti, Intel Core i9-10920X and M1 Pro. With GAN Compression, we could further speedup GauGAN by 4.0× on Intel Core-i9 and 11× on Apple M1 Pro.

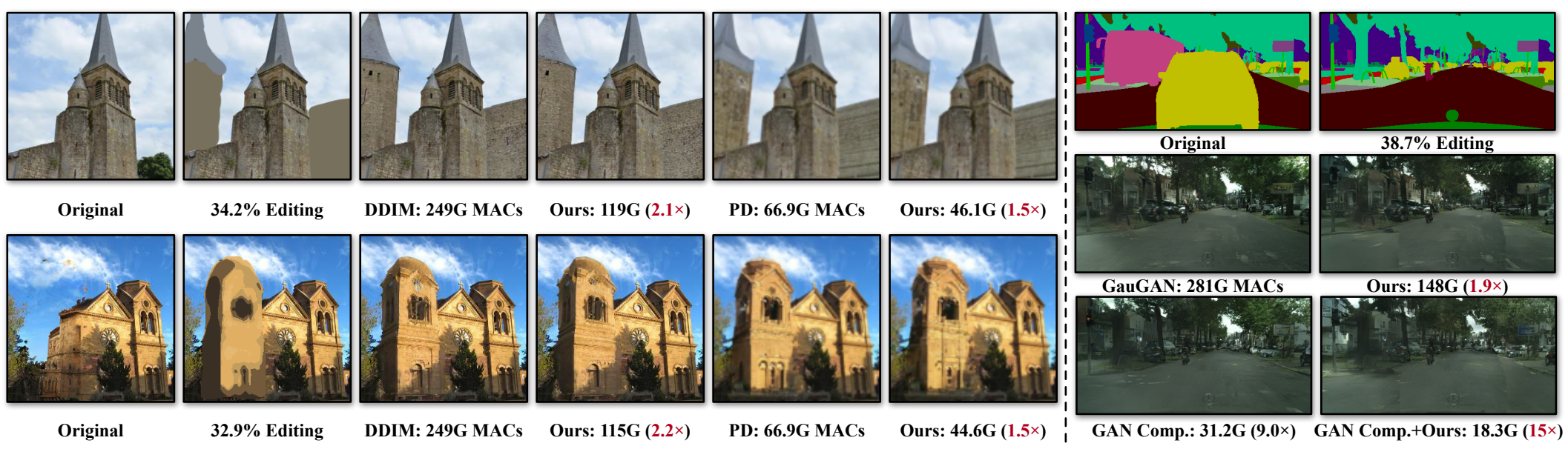

Figure 11: Qualitative results of our method under large editing. Our method could still well preserve the visual fidelity of the original model without losing global context while reducing the computation by $1.5 \sim 1.9\times$.

Since our method does not involve any model training, all our generated results are obtained on a single NVIDIA RTX 3090, which only takes $1 \sim 2$ hours to process all the test images ($\sim 7,000$ in total) including both the original models and our method. We measure the model latency on NVIDIA RTX 3090, 2080Ti, Intel Core i9-10920X CPU, and Apple M1 Pro.

# F Discussion

**Limitations.** As discussed in Section 4.2, our method needs some additional memory to store the original activations, even though this only increases the peak GPU memory usage slightly. It may not work on some memory-constrained devices, especially for the diffusion models (*e.g.*, DDIM [1] and Progressive Distillation [12]), since our method requires storing activations of all iteration steps.

Our engine has limited speedup on convolution with low resolution. When the input resolution is low, the sparse block size needs to be even smaller to get a good sparsity, such as 1 or 2. However, such extremely small block sizes have worse memory locality and will result in low hardware efficiency.

Besides, we sometimes observe noticeable boundary between the edited region and unedited region in our generated samples of GauGAN [2]. This is because, for GauGAN model, the unedited region will also change slightly when we perform normal inference. However, since our method does not update the unedited region, there may be some color gaps between the edited and unedited region, even though the semantic is coherent. Dilating the difference mask would help reduce the gap.

**Societal impact.** In this paper, we investigate how to update user editing locally without losing global coherence to enable smoother interaction with the generative models. In real-world scenarios, people could use an interactive interface to edit an image, and our method could provide a quick and high-quality preview for their editing, which eases the process of visual content creation and saves energy.

However, our method can also be utilized by some malicious users to generate fake content, deceive people, and spread misinformation, which may lead to potential negative social impacts. Following previous work [9], we will also explicitly specify the usage permission of our engine with proper licenses.

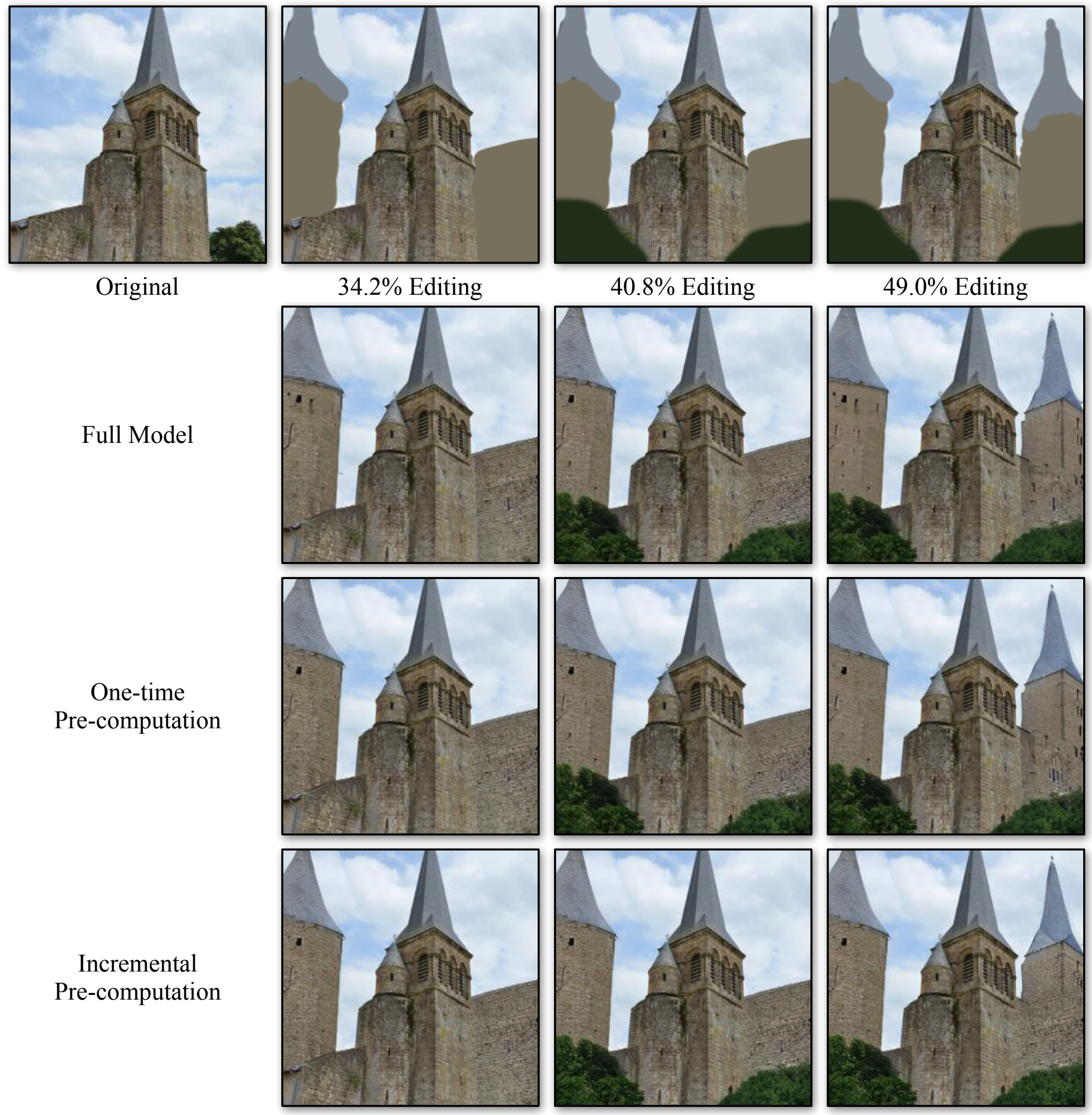

Figure 12: Sequential editing results with SIGE. *Full Model* means the results with the full model. *One-time Pre-computation* means we only pre-compute the original image features for all the editing steps. *Incremental Pre-computation* means we incrementally update the pre-computed features with SIGE before the next editing step. The image quality of all methods are quite similar.