# OpenReview forum: "Efficient Spatially Sparse Inference for Conditional GANs and Diffusion Models"
_NeurIPS.cc/2022/Conference — NeurIPS 2022 Accept_

### Official Review · Reviewer_oeJz · 2022-06-22

**Rating:** 4
**Confidence:** 4
**Soundness:** 3 good
**Presentation:** 3 good
**Contribution:** 2 fair

**Summary:**

This paper proposed the Sparse Incremental Generative Engine (SIGE), which consists of several tricks (tiling-based sparse convolution, the use of pre-computing normalization parameters, and kernel fusion) to accelerate local image edits of generative models with little sacrifice in image quality.

**Questions:**

Please address the comments in the Weakness section.

**Limitations:**

Please see the Weakness section.

**Strengths And Weaknesses:**

Strength:
- (Clarification) This paper is well-written and easy to follow.
- (Quality) The evaluation of latency reduction on various hardware is interesting and useful.

Weakness:
- (Originality) This paper is a bit incremental and engineering: i) the key insight is straightforward and not surprising as the spatial correspondences between features of convolutional layers and generated images are well-known. ii) the proposed three tricks in SIGE are mostly engineering ones and a bit thin. Although they are given good names, they are a bit trivial without significant technical contributions, e.g. the core idea of tiling-based sparse convolution is mostly borrowed from SBNet with an incremental improvement trick, the use of pre-computing normalization parameters is a simple approximation that can be explained in a single sentence, the kernel fusion is mostly engineering tricks that only perform computation in the edited region and the copying overhead.
- (Significance) The significance of the proposed method could be less than expected as it has an important limitation: its benefits may not hold for overlapping editing regions in a sequence of edits, which is a common case in the real world. In this case, the pre-computation stage is no longer one-time (i.e. the generated image may need to be pre-computed again before the next edit) and can make the proposed method less useful.
- (Quality) The dilation hyper-parameters could be important to the final results and should be discussed in more depth.

Missing References:

For "Generative models" in section 2, there are missing references for image-to-image translation and real image editing:

[1] Abdal, R., Qin, Y. and Wonka, P., 2019. Image2stylegan: How to embed images into the stylegan latent space?. In Proceedings of the IEEE/CVF International Conference on Computer Vision (pp. 4432-4441).

[2] Abdal, R., Qin, Y. and Wonka, P., 2020. Image2stylegan++: How to edit the embedded images?. In Proceedings of the IEEE/CVF conference on computer vision and pattern recognition (pp. 8296-8305).

[3] Zhu, P., Abdal, R., Qin, Y. and Wonka, P., 2020. Sean: Image synthesis with semantic region-adaptive normalization. In Proceedings of the IEEE/CVF Conference on Computer Vision and Pattern Recognition (pp. 5104-5113).

Minor/Typo:
- Line 131: missing "."
- Table 1 caption: "LIPS" -> LPIPS
- In Table 1, the bold fonts are not always the best results, e. g. GAN Comp. LPIPS with G.T. is better than the proposed method.

---

> ### Author Response · Authors · 2022-08-02
> **Author Response to Reviewer oeJz**
>
> ### Originality
> We respectfully disagree with the review’s opinions that our paper is incremental and engineering. In our general response, we clarify and highlight the novelty of our algorithm and engine.
> * Our key insight is NOT the spatial correspondence between the features of convolutional layers and generated images. Our main insight is to selectively update the local edited regions instead of the whole image to accelerate generative image editing. To achieve this, we pre-compute the feature maps and reuse them for updating the edited regions sparsely. To the best of our knowledge, this is a new contribution to efficient deep image editing, which can be applied to a wide range of generative models (e.g., GANs and diffusion models) and could be of great interest to a broad audience.
> * Our engine optimizations are non-trivial and not just engineering. They are co-designed with our algorithm to better adapt the SBNet for generative image editing. Directly using SBNet as our engine yields poor performance, as it was originally designed for recognition tasks. As shown in Table 3 of the paper, it only has a $1.6\times$ speedup even though the computation reduction is $9.1\times$. Our normalization parameter pre-computation also reuses the original image results from our algorithm and enables us for further kernel fusion. With these optimizations, we achieve $2.9\times$ latency reduction, which is $1.8\times$ faster than the original SBNet. This shows that our engine optimizations contribute more significantly to our final performance.
>
> ### Sequential editing does not dilute our contribution
> Thanks for the discussion. Our method can be applied to multiple sequential editings with only **one-time** pre-computation. In many cases, the editing will not change the global context (e.g., adding two trees with overlaps). Therefore, our method could speed up updating all the edited regions with the context of original images while not losing visual fidelity using just a one-time feature pre-computation.
>
> Moreover, we can further perform full feature recomputation during the idle time (happens a lot in a real editing setting (e.g., AnyCost GAN [59])) to reset the error accumulations and condition the later editing on the current feature maps, which can lead to a better speed up and quality for future editings.
>
>
> ### Dilation hyper-parameters
> As suggested by the reviewer, we show the results of different dilation sizes on GauGAN in this [figure](https://anonymous.4open.science/r/sige-neurips22-rebuttal-937C/dilation.png) (or Figure 10 in our revised supplementary materials). Large dilation could slightly improve the image quality by smoothly blending the edited and unedited regions at the cost of  extra computation. Specifically, the shadow boundary of the added car fades when the dilation is 20. We choose dilation 1 in our experiments since the image quality is almost the same as 20 while delivering the best speed. We have included the results in our revision (see Section D and Figure 10 in the supplementary materials).
>
> ### Reference
>
> Thanks for pointing out the references. We have cited these works in our revision (see Section 2 Generative Models).
>
> ### Minor & typos
>
> Thanks for pointing them out. We have fixed them in our revisions.
>
> We hope our response has resolved all of your concerns. Please let us know what other experiments or clarifications we can offer to convince you to increase the rating.

---

> > ### Author Response · Authors · 2022-08-08
> > **Gentle Reminder**
> >
> > Thanks again for your insightful comments.
> > In our previous response, we have added additional experiments and analyses accordingly to your suggestions. Please do not hesitate to contact us if you have additional questions.  Thank you for your time again!
> >
> > Best, Authors

---

> > > ### Comment · Reviewer_oeJz · 2022-08-08
> > > **Reply to authors**
> > >
> > > Thank the authors for their reply. My concerns on dilation are resolved but I am still not fully convinced by the originality and sequential editing part of the paper. I hope to see other reviewers' opinions in the discussion period before making the final decision.

---

> > > > ### Author Response · Authors · 2022-08-09
> > > > **Author Response**
> > > >
> > > > Thanks for your additional comments. In our general response, we highlight our novelty and differences between our work and SBNet. We also include more results to demonstrate that our method works well for sequential editing. Please do not hesitate to contact us if you have additional questions. Thank you for your time again!

---

### Official Review · Reviewer_ckKG · 2022-06-26

**Rating:** 6
**Confidence:** 4
**Soundness:** 4 excellent
**Presentation:** 3 good
**Contribution:** 3 good

**Summary:**

This paper proposes spatially sparse inference (SSI) to accelerate interactive generative image editing. The key idea is to keep a backup of the original features and only perform computation for edited regions, which is usually sparse in the interactive setting. The paper also introduces some implementation improvements to further reduce the computational overhead, including pre-computing normalization parameters and kernel fusion. Experiments on several pipelines, datasets, and devices show that the proposed method can drastically reduce the computation and running speed, and outperforms other pruning or patch-based baselines. Ablation studies are conducted to verify the effectiveness of each proposed module.

**Questions:**

1. Considering that the key idea of spatially sparse inference is not new and that the key spatially sparse computing module is based on SBNet [67], the novelty of the proposed method is not significant to me. Authors may highlight their novelty.

2. Please explain why $\Delta A_l$ is divided into blocks instead of pixels (line 130).

3. Do the "patch" and "crop" baselines use pre-computing normalization and kernel fusion? Please clarify this.

**Limitations:**

Limitations and potential negative societal impact have been properly discussed.

**Strengths And Weaknesses:**

Strengths:

1. This paper takes the first attempt to apply region-specific computation to generative image editing. It makes use of the sparse nature of interactive image editing and avoids redundant computation in the unedited regions, which is very reasonable. As interactive image editing is a broadly studied application scenario and the running speed is important, I believe this work is of high interest to the community.

2. To improve practical computing efficiency, the paper proposes pre-computing normalization parameters and kernel fusion. These implementation improvements effectively reduce computational overhead and accelerate inference in practice.

3. Experimental studies are thorough. The proposed method is evaluated on several different pipelines, datasets, and devices, and its advantages over baselines are clearly demonstrated.

4. The writing is generally clear and easy to follow.

Weaknesses:

1. The main weakness is the novelty, as the key idea of spatially sparse inference is not new and has been widely used before, e.g., [67, 72, 73, 74]. There is also a missing related work that performs region-specific computation: "Not All Pixels Are Equal: Difficulty-Aware Semantic Segmentation via Deep Layer Cascade". In particular, the proposed method is mainly built based on SBNet [67]'s sparse kernel implementation. The differences in terms of how mask is derived and other implementation improvements are not very significant.

The following weaknesses are minor:

2. At line 130, it is not explained why $\Delta A_l$ is divided into blocks instead of pixels.

3. Do the "patch" and "crop" baselines use pre-computing normalization and kernel fusion? Please clarify this.

Typos:
Line 251: could saves -> could save
Line 262: the activations -> activations

---

> ### Author Response · Authors · 2022-08-02
> **Author Response to Reviewer ckKG**
>
> ### Novelty
> In our general response, we highlight the contribution and novelty of our algorithm and engine. Our key insight is that the original and edited images are quite similar. Therefore, we can reuse the original image results to accelerate deep generative image editing. We believe this high-level idea is a new contribution to the field of generative models. To implement this idea, we leverage the spatial sparsity inside the edited image when reusing the original image feature maps. Directly using existing tiling-based sparsity methods such as SBNet only yields small gains, as it was primarily designed for recognition tasks.  On the contrary, our algorithm-engine co-design is tailored for generative models and outperforms SBNet by $1.8\times$.
>
> ### Why $\Delta A_l$ is divided into blocks:
> This is because a $3 \times 3$ convolution needs to operate on at least a $3 \times 3$ region, so we could not divide $\Delta A_l$ into pixels. For $3\times3$convolution, we include additional overlaps for  the divided blocks to ensure that the output blocks of the adjacent input blocks can be seamlessly stitched together.
>
> ### Optimizing baselines
> Thanks for your suggestion.  The "patch" and "crop" baselines did not use pre-computing normalization and kernel fusion in the original paper. We did not optimize these baselines, as they didn’t work well or generalize to arbitrary editing regions. The image quality of "Patch" baseline already degrades for small editing regions. The "Crop" baseline incurs redundant computations for irregular regions (e.g., the third cloud example in Figure 5).
> Following reviewers’ suggestions, we further optimized the "Crop" baseline by adopting pre-computed normalization (see the following table or Table 2 in the revised paper). We didn’t use kernel fusion as the crop baseline is based on highly-optimized PyTorch implementation. Our method still consistently outperforms this optimized baseline, especially when the editing is irregular (e.g., the third cloud example in Figure 5). We have clarified and updated the results in our revision (see Table 5 and Line 257-259).
>
>
> | Editing Size | Method   | MACs                | 3090         | 2080Ti       | Intel Core i9-10920X | Apple M1 Pro |
> | :----------- | -------- | ------------------- | -------------------- | -------------------- | ---------------------------- | -------------------- |
> | --           | Original | 248G                | 37.5ms               | 54.6ms               | 609ms                        | 12.9s                |
> | 1.20%        | Crop     | 32.6G (7.6$\times$) | 15.5ms (2.4$\times$) | 29.3ms (1.9$\times$) | 185ms (3.3$\times$)          | 1.85s (6.9$\times$)  |
> |              | Ours     | 33.4G (7.5$\times$) | 12.6ms (3.0$\times$) | 19.1ms (2.9$\times$) | 147ms (4.1$\times$)          | 1.96s (6.6$\times$)  |
> | 7.19%        | Crop     | 54.4G (4.6$\times$) | 17.3ms (2.2$\times$) | 26.5ms (2.1$\times$) | 220ms (2.8$\times$)          | 2.98s (4.3$\times$)  |
> |              | Ours     | 51.8G (4.8$\times$) | 15.5ms (2.4$\times$) | 22.1ms (2.5$\times$) | 223ms (2.7$\times$)          | 3.23s (4.0$\times$)  |
> | 15.5%        | Crop     | 155G (1.6$\times$)  | 30.5ms (1.2$\times$) | 44.5ms (1.2$\times$) | 441ms (1.4$\times$)          | 8.09s (1.6$\times$)  |
> |              | Ours     | 78.9G (3.2$\times$)  | 19.4ms (1.9$\times$) | 29.8ms (1.8$\times$) | 304ms (2.0$\times$)          | 5.04s (2.6$\times$)  |
>
> ### Reference
> Thanks for pointing out the reference. We have cited this work and discussed it in our revision (see Section 2 Sparse Computation).
>
> ### Typos
> Thanks for pointing them out. We have revised them accordingly.
>
> We hope our response has resolved all of your concerns. Please let us know what other experiments or clarifications we can offer to convince you to increase the rating.

---

### Official Review · Reviewer_2N5G · 2022-07-12

**Rating:** 5
**Confidence:** 3
**Soundness:** 2 fair
**Presentation:** 2 fair
**Contribution:** 2 fair

**Summary:**

This paper produces a speedup technique, spatially sparse inference, for the image manipulation method. The SIGE significantly reduces inference time and does not harm the performance of the original network.

**Questions:**

1. How about performance when editing regions are large(over 30%)?
2. How to get the correct index of editing regions. In my opinion, since the convolution process, the difference mask  in Fig.3 can not represent the editing regions in the deep level of network.

**Limitations:**

See the questions.

**Strengths And Weaknesses:**

Strengths:
1. The motivation is reasonable. Finding the edit regions and making the inference process focus on the edit regions, this strategy can reduce the time cost intuitively.
2. The experiments are sufficient. Table 2 can prove the ability of SSI well.

Weaknesses:
See the questions.

---

> ### Author Response · Authors · 2022-08-02
> **Author Response to Reviewer 2N5G**
>
> ### Effectiveness for large editing
> As suggested by the reviewer, we show the results of large editing (around 35%) in the following table and this [figure](https://anonymous.4open.science/r/sige-neurips22-rebuttal-937C/large-editing.png) (or Table 4 and Figure 11 in our revised supplementary materials). Specifically, we could achieve up to $1.7 \times$ speedup on DDIM, $1.5 \times$ speedup on PD256, and $1.7 \times$ speedup on GauGAN without losing visual fidelity. Furthermore, in many practical cases, users can decompose a large edit into several small ones. Our method could incrementally update the results when the edit is being created. We have included the results in our revision (see Section D, Table 4, and Figure 11 in the supplementary materials).
>
> | Model  | Editing Size | Method         | MACs                | 3090                 | 2080Ti               | Intel Core i9 | Apple M1 Pro        |
> | ------ | ------------ | -------------- | ------------------- | -------------------- | -------------------- | -------------------- | ------------------- |
> | DDIM   | --           | Original       | 249G                | 37.5ms               | 54.6ms               | 609ms                | 12.9s               |
> |        | 32.9%        | Ours           | 115G (2.2$\times$)  | 26.0ms (1.4$\times$) | 36.9ms (1.5$\times$) | 449ms (1.4$\times$)  | 7.53s (1.7$\times$) |
> | PD256  | --           | Original       | 119G                | 35.1ms               | 51.2ms               | 388ms                | 6.18s               |
> |        | 32.9%        | Ours           | 64.3G (1.9$\times$) | 25.3ms (1.4$\times$) | 35.1ms (1.5$\times$) | 334ms (1.2$\times$)  | 4.47s (1.4$\times$) |
> | GauGAN | --           | Original       | 281G                | 45.4ms               | 49.5ms               | 682ms                | 14.1s               |
> |        | --           | GAN Comp.      | 31.2G (9.0$\times$) | 17.0ms (2.7$\times$) | 25.0ms (2.0$\times$) | 333ms (2.1$\times$)  | 2.11s (6.7$\times$) |
> |        | 38.7%        | Ours           | 148G (1.9$\times$)  | 27.9ms (1.6$\times$) | 41.7ms (1.2$\times$) | 512ms (1.3$\times$)  | 8.37s (1.7$\times$) |
> |        | 38.7%        | GAN Comp.+Ours | 18.3G (15$\times$)  | 15.3ms (3.0$\times$) | 22.2ms (2.2$\times$) | 169ms (4.0$\times$)  | 1.25s (11$\times$)  |
>
> ### Edited region indexing
> Thanks for your question. We downsample the difference mask to different resolutions and dilate the downsampled mask with extra pixels (1 for diffusion models and 2 for GauGAN). For each convolution inside the network, we use the difference mask at the corresponding resolution to index the active blocks. For $3 \times 3$ convolution, we include extra overlaps (2) for the indexed active blocks to ensure that  the output blocks of the adjacent input blocks can be  seamlessly stitched together. We don’t use overlaps for $1\times1$ convolutions.  We have clarified this in our revision (see Section 3.2).
>
> We hope our response has resolved all of your concerns. Please let us know what other experiments or clarifications we can offer to convince you to increase the rating.

---

> > ### Comment · Reviewer_2N5G · 2022-08-08
> > **Response to Authors**
> >
> > I appreciate that the authors took the time to carry out the analysis I suggested. I think the authors answered my question well. But I read other reviewers' questions, and I find that the method is similar to the SBNet. I will participate in the discussion and give the final decision.

---

> > > ### Author Response · Authors · 2022-08-09
> > > **Author Response**
> > >
> > > Thanks for your additional comments. In our general response, we highlight our novelty and differences between our work and SBNet. Please do not hesitate to contact us if you have additional questions. Thank you for your time again!

---

### Author Response · Authors · 2022-08-02
**General Response to All Reviewers and ACs**

We sincerely appreciate all reviewers' efforts for the insightful and thoughtful comments. We are glad that the reviewer recognized the following strengths.

* Motivation & Contribution: The motivation of reusing the original image activations to update the sparse edited regions is reasonable and intuitive (reviewer 2N5G and ckKG). Our work is of high interest to the community (reviewer ckKG).
* Experiments: The experimental studies are thorough across different models, datasets, and devices to show the method’s effectiveness (reviewer 2N5G, ckKG, and oeJz).
* Presentations: The paper writing is generally clear and easy to follow (reviewer ckKG and oeJz).

In addition to the pointwise responses below, we first clarify and emphasize our contribution and novelty and then summarize the major changes  in our revision:

1. Contribution & novelty
   * Our high-level idea is to selectively update the local edited regions instead of the whole image to accelerate generative image editing. To achieve this, we pre-compute the feature maps and reuse them for updating the edited regions sparsely. To the best of our knowledge, this is a new contribution to efficient deep image editing, which can be applied to a wide range of generative models (e.g., GANs and diffusion models) and could be of great interest to a broad audience.
   * To achieve our high-level idea, we propose engine optimizations that are tailored for generative models and co-designed with our algorithm. Directly applying existing techniques (e.g., SBNet) fails to yield significant speedup. In contrast, our engine optimization significantly outperforms SBNet by $1.8\times$.

2. Revision summary.
  We made the following revisions to our manuscript to address the reviewers’ comments:
   * As suggested by reviewer oeJz, we include the ablation of different dilation sizes in Section D of the supplementary materials. Large dilation could slightly improve the image quality by smoothly blending the edited and unedited regions at the cost of  extra computation.
   * As suggested by reviewer 2N5G, we include the results of large editing in Section D of the supplementary materials. Our method could achieve up to $1.7\times$ speedup without losing visual fidelity for the  $\sim35$% editing.
   * As suggested by reviewer ckKG, we clarify the experimental setting and update the results of the “Crop” baseline in Section 4.1. Our method could still beat the optimized baseline.
   * As suggested by reviewer ckKG and oeJz, we include and discuss the additional related works in Section 2.

---

> ### Author Response · Authors · 2022-08-08
> **Gentle Reminder**
>
> Dear AC and all reviewers:
>
> Thanks again for all of your constructive suggestions, which have helped us improve the quality and clarity of the paper. Please don’t hesitate to let us know if there are any additional clarifications or experiments that we can provide.
>
> Best, Authors

---

> > ### Author Response · Authors · 2022-08-09
> > **General Response**
> >
> > ## Novelty
> >
> > Thanks for your additional comments. We highlight our novelty and differences between our work and SBNet as follows:
> >
> > * High-level ideas are different: Our novelty is **NOT** to propose a new tiling-based sparse convolution but provide a new technique for accelerating deep generative image editing. Our key idea is to reuse the original image features to update the edited images as they are quite similar. To the best of our knowledge, previous work on generative model acceleration mainly focuses on reducing model sizes (e.g., channel pruning and quantization), but few of them explore the sparsity inside the activations. We are the first to uncover the spatial sparsity between the original and edited features and leverage it for further speedup.
> > * Performances are different. Directly using SBNet yields poor performance (only $1.6\times$ faster than the baseline) as it mainly targets recognition networks. Our engine is co-designed with the generative networks and our algorithm and is $1.8\times$ faster than SBNet and $2.9\times$ faster than the baseline.
> > * Applications are different: Our work tackles interactive editing with generative models, while original SBNet was used in 3D object detection.
> >
> > ## Our method works well for sequential editing
> >
> > Thanks for your additional comments. In this [figure](https://ibb.co/b5f3Cdg) (or Figure 12 in our revised supplementary materials), we show the results of sequential editing with the following methods:
> >
> > * Full Model: the results with the full model.
> > * One-time pre-computation: we only pre-compute the original image features for all the editing steps.
> > * Incremental pre-computation: we incrementally update the pre-computed features with SIGE before the next editing step.
> >
> > Specifically, One-time pre-computation performs as well as the full model, demonstrating that our method can be applied to multiple sequential editing with only one-time pre-computation in most cases. Moreover, for extremely large edited regions, we could use SIGE to incrementally update the pre-computed features (Incremental Pre-computation) and condition the later editing on the recomputed one. Its results are also as good as the full model. Therefore, our method could well address the sequential editing. We have included the results in our revision (see Section D and Figure 12 in the supplementary materials).

---

### Meta-Review · Area_Chair_7S2L · 2022-08-27

**Recommendation:** Accept
**Confidence:** Less certain

**Metareview:**

This paper was a close call. One reviewer was of the opinion that the paper lacked significant innovations other than fairly obvious sparse processing tricks to make local edits faster. This reviewer did not change their opinion (Borderline Reject) post-rebuttal. Of the other two reviewers, one was at Borderline Accept and the third at weak accept. After reading the paper, I agree with some of the first reviewer's comments on a number of technically-obvious contributions. However, I also believe these contributions are valuable from a practical perspective. Furthermore, code release (as promised by the authors) will be valuable to the community. Therefore, I recommend acceptance.

**Award:**

No

---

### Decision · Program_Chairs · 2022-09-14

Accept